RESEARCH CULTURE

# The SAFE Labs Handbook as a tool for improving lab culture

**Abstract** Creating positive and equitable lab environments has become a growing priority for the scientific community and funders of scientific research. Research institutions typically respond to this need by providing mandatory or optional training opportunities for their staff. However, there are limited resources for group leaders to improve the culture in their labs. Here, we introduce the SAFE Labs Handbook: a collection of 30 "commitments" that can be implemented by individual group leaders to improve research culture in the life sciences. The commitments were collaboratively developed by 13 group leaders working in eight different European countries. We also report the results of a survey in which we asked more than 200 researchers, at various career stages, about the commitments. Even though all 30 commitments were rated as significantly important by respondents, implementation rates were notably low (<25%). However, more than 95% of group leaders said they would consider implementing them. The SAFE Labs Handbook therefore represents a unique, community-driven tool with the potential to improve lab culture on a global scale.

**ERIKA DONÀ[†], JAMES M GAHAN[†], PETRINA LAU[†], JANA JESCHKE[†], TORBEN OTT[†], KATJA REINHARD[†], CHIARA SINIGAGLIA[†], JORIEN L TREUR[†], THOMAS VOGL[†], STEPHANE BUGEON*[‡], LETIZIA MARIOTTI*[‡], L FEDERICO ROSSI*[‡], PHILIP COEN*[‡]**

**\*For correspondence:**
stephane.bugeon@inserm.fr (SB);
letizia.mariotti@cnr.it (LM);
federico.rossi@iit.it (LFR);
p.coen@ucl.ac.uk (PC)

[†]These authors contributed equally to this work
[‡]These authors also contributed equally to this work

**Competing interest:** The authors declare that no competing interests exist.

## Introduction

The importance of creating a positive, fair and transparent working environment in academia has become increasingly evident over the past decade. Cross-sectional surveys and institutional audits reveal high self-reported levels of stress, poor work-life balance, and disparities in pay and career progression (*Acton et al., 2019*; *Evans et al., 2018*; *Guthrie et al., 2018*; *Kim et al., 2024*; *Kinman and Jones, 2008*; *Marck et al., 2024*; *Morgan et al., 2021*; *Susi et al., 2019*; *Wellcome Trust, 2020*). While these studies lack demographically matched non-academic control groups, they identify systemic issues within academia that need to be addressed. Solving these problems requires intervention at all levels of the academic hierarchy. Making top-down changes at the national or institutional level – such as those proposed in the Researcher Development Concordat in the UK (*RDC Strategy Group, 2019*) – can take time. Conversely, bottom-up changes made by individuals – such as those proposed in this article – can be implemented more rapidly.

The research group (or "lab") is the foundational unit throughout the academic world, and the policies of the group can determine the well-being of the people who work in it (*Hammoudi Halat et al., 2023*). The group leader is typically responsible for determining the policies of their group, and hence has the strongest influence on the lab culture. There are informative resources with advice for starting a research group (*Aly, 2018*; *Goldstein and Avasthi, 2021*; *Schmidt, 2006*; *Somerville et al., 2019*), and a number of groups have made the lab handbooks and manuals that document their various policies publicly available (*Aly, 2018*; *Andreev et al., 2022*; *Martin and Stanfill, 2023*; *Tendler et al., 2023*). In addition to being useful to people who are already part of the group, public handbooks and manuals are also helpful to individuals who are thinking of joining it.

In this article, we take a different approach with the SAFE Labs Handbook, creating a community-driven document that is applicable to any field of research within the life sciences. The handbook comprises 30 actions (or "commitments") that can be taken to improve the culture in a research group. These commitments are

all actionable items that can be directly implemented by a group leader, and they all contain well-defined, measurable outcomes. Moreover, the commitments can be implemented without institutional support. We also report the results of a survey in which more than 200 researchers from more than 20 countries – including group leaders, postdoctoral researchers, PhD students, and staff – were asked about the 30 commitments. The complete handbook, explanations of each commitment, and a growing collection of templates from international groups across disciplines, can be found here: https://safelabs.info/home/safe-labs-handbook/.

## The SAFE Labs project

This article has its origins in the Starting Aware, Fair, and Equitable Labs (SAFE Labs) workshop that was attended by the present authors – all new group leaders in the life sciences – in Palazzone di Cortona, Italy in May 2024. The workshop was organized to explore how to foster more positive and equitable lab environments. Although cultural problems in academia are wide-ranging, the workshop focused on how group leaders could improve their lab environment without institutional support. The feedback on the workshop from the attendees was overwhelmingly positive, with everyone saying that they would make changes in their labs as a result of attending the workshop (*Figure 2—figure supplement 1*).

The workshop highlighted two key barriers to improving lab culture as a new group leader. First, existing resources provide either general advice or example handbooks from specific labs, rather than actionable steps that can be implemented in verifiable manner. Second, academia does not normalise the documentation of lab policy, with group leaders often relying on *word-of-mouth* to educate new lab members. When documentation does exist, key information is often limited to existing lab members, and is not available to prospective applicants.

Attendees at the workshop agreed that the "best" lab environment was dependent on the individual. However, they also agreed that all groups could benefit from documenting key policies to minimise expectation mismatch between lab members (existing and prospective) and the group leader. To achieve this goal, attendees agree to create a handbook to: (i) provide actionable steps for group leaders to improve their lab environments without institutional support; (ii)

normalize the process of documenting key lab policies.

While the initial workshop targeted new group leaders to ensure equality of experience among attendees, and to encourage discussion, the commitments in the SAFE Labs Handbook are relevant to all research groups, irrespective of how experienced the group leader is. Moreover, subsequent workshops have been open to all applicants.

### The SAFE Labs Handbook

The SAFE Labs Handbook comprises 30 commitments spanning three broad categories: Teams, Policies, and Careers (*Figure 1*). Each commitment is an actionable statement that asks a group leader to publicly document, internally document, or establish a policy in their laboratory. Publicly documented information should be visible to anyone (e.g. posted on the lab website). Internally documented information need only be visible to existing lab members (e.g. posted in a password-protected lab manual or wiki): this may be information that is sensitive, or only relevant to individuals who are current members of the lab. In this context "establish" indicates a commitment that requires a change at the level of lab management, such as organizing a new annual meeting. Commitments were generated through a distillation of the notes from the workshop by the 13 attendees (see Methods). Publicly vs internally documented commitments were established through anonymous vote, with the decision deferred to the community-survey if there was no consensus (10%<score < 50% in favour, *Figure 2—figure supplement 1h*).

Commitments were chosen to be actionable and verifiable to facilitate implementation and accountability. Each commitment derived from collated documentation from the SAFE Labs Workshop. We identified prominent issues that warranted inclusion in the handbook. For example: expectations around working hours and vacation are unclear to prospective and current lab members. We then agreed on the form of the commitment that made it actionable, verifiable and minimally prescriptive. "I commit to supporting a healthy work-life balance" would *not* qualify because the commitment cannot be verifiably implemented. Instead, the analogous entry in the handbook states that a group leader commits to "publicly document expectations for working hours, remote working, and vacation." This is actionable (requires specific action from the group leader), verifiable (implementation can

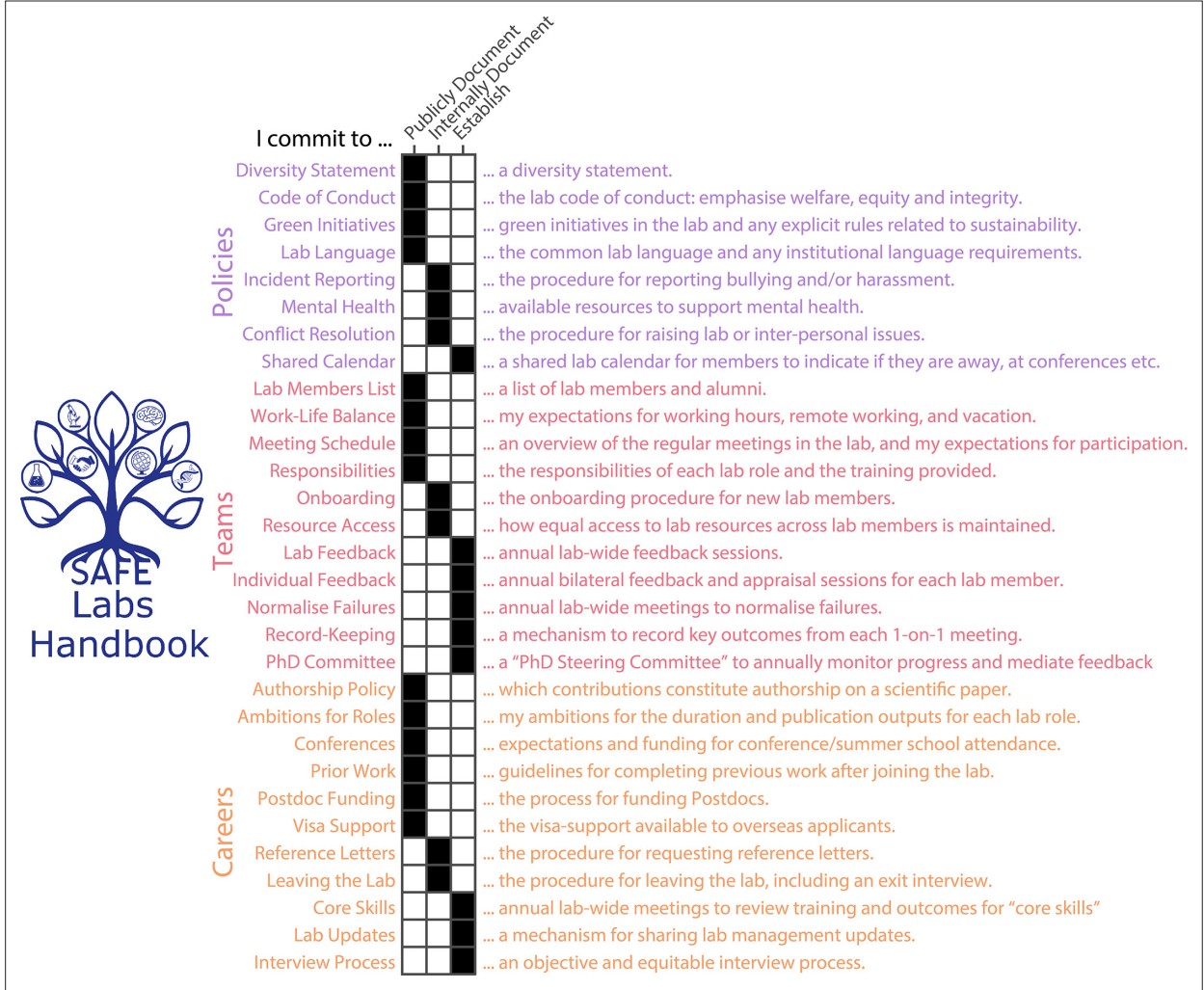

**Figure 1.** The commitments in the SAFE Labs Handbook. The SAFE Labs Handbook logo (left) and summary of its commitments (right), arranged by category. Matrix indicates (black) whether each commitment should be publicly documented, internally documented, or established. The complete handbook, along with explanations of each commitment and example text, can be found here: https://safelabs.info/home/safe-labs-handbook/.

be evidenced), and does not mandate any particular policy – it instead requires that the policy is documented so that expectations are clear to all current and prospective lab members. We therefore provide only details, suggestions, and where appropriate, a freely editable template statement for each commitment. We also encourage the growing community to add examples of their statements, to help new group leaders develop effective policies tailored to their environment.

As an illustration, here is the current entry for the above commitment:

*Details*
*Lab rules for working hours should be clear to avoid conflicts or misunderstandings, and clear expectations for working hours can increase equity between lab members. Group leaders*

*need to ensure that lab members feel safe to balance their work in the lab with their life outside it. Policies regarding work hours, remote working, and vacation should be explicitly included.*

*Suggestions*
*Should notice of holidays be given and how? Are there core-working hours (typically less than the full working hours)? Should lab members schedule messages if sent outside of working hours? What times are appropriate times for scheduling meetings?*

*Template*
*I am committed to creating a healthy work environment for all lab members. I anticipate all lab members taking a minimum of [institution's]*

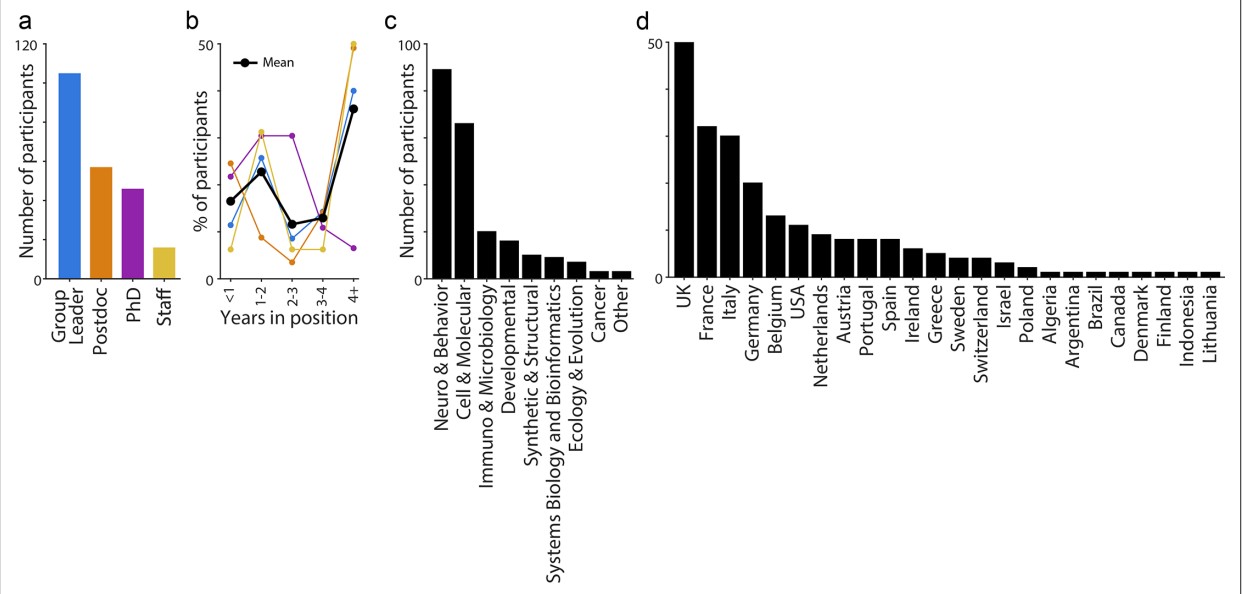

**Figure 2.** Survey: demographics of respondents. (**a**) Number of respondents to the SAFE Labs Handbook survey, sorted by employment role. (**b**) Fraction of respondents as a function of seniority in each role from (a, color). Black line indicates the average across all respondents. (**c**) Number of respondents sorted by research field. (**d**) As in (**c**), but by country of employment. For all panels, n=105 group leaders, 57 postdocs, 46 PhD students, and 16 staff.

The online version of this article includes the following figure supplement(s) for figure 2:

**Figure supplement 1.** 2024 SAFE Labs Workshop demographics and feedback.

*prescribed days of annual leave. "Minimum" because if experiments/conferences necessitate working on a weekend, I support lab members taking time off to compensate. I hope to schedule all meetings within [institution's] "core" work hours, and will refrain from sending, or answering, non-urgent emails/messages outside of work hours.*

*Full-time lab members should work "onsite" at least four days a week. I believe regular onsite presence is important to maintain the lab community. However, I will support intermittent periods of remote work when, for example, traveling/ visiting family abroad or writing up a thesis/grant.*

## Results of a community survey

To assess the need for each commitment, we conducted a survey in which over 200 researchers were asked to rate the importance of every commitment, and to state whether it was already implemented in their environment. The number of respondents in the "group leader" career stage (51%, n=105), suggests a strong drive to improve lab culture at the senior level. The remaining 49% of respondents were split amongst postdocs (25%, n=57), PhD students (21%, n=46), and research support staff (7%, n=16; *Figure 2a*). Further, more than 30% of respondents (over

40% of non-PhD student respondents) had been in their role for over four years – demonstrating that both new and experienced members of the community engaged with our survey (*Figure 2b*). Participants represented institutions in 24 countries and worked in a range of fields (*Figure 2c and d*). Although these demographics are influenced by organiser bias, no single country represented more than 25% of participants. Overall, participant demographics indicate broad engagement with the commitments in the SAFE Labs Handbook.

When asked to rank the importance of the commitments on a scale of 1–5 – where 1 was "useless" and 5 was "critical" – the average response was around 4 for all career stages (*Figure 3a and b*) and countries (*Figure 3— figure supplement 1a–d*). However, the average implementation rate for the commitments was only about 25% (*Figure 3c and d*). The mean importance score amongst respondents was not significantly impacted by their years of experience (*Figure 3—figure supplement 2*). Across the five countries with the most respondents, the relative importance of commitments and implementation rates were broadly consistent, with the UK and Italy reporting the highest and lowest implementation rates (*Figure 3—figure*

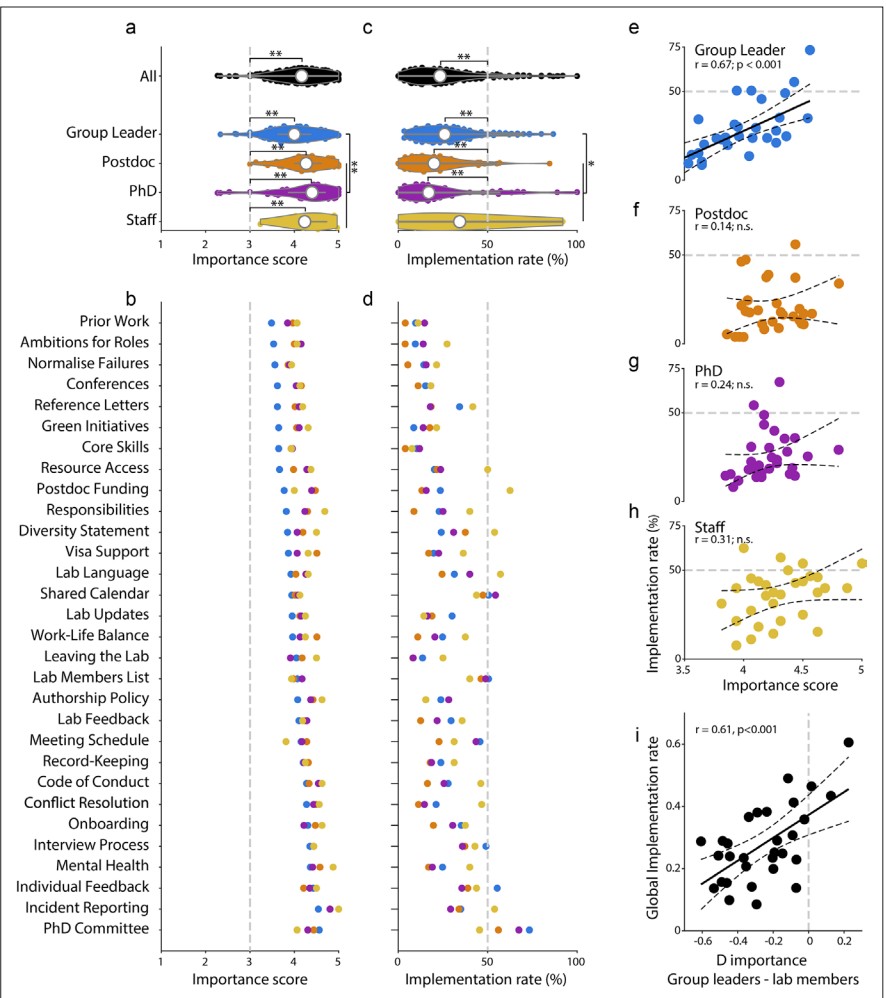

**Figure 3.** Survey: different response from group leaders and lab members. (**a**) Distribution of mean importance score across commitments, for all survey respondents (black), and for each career stage: group leader (blue), postdoc (orange), PhD student (purple), and staff (yellow). Violin plot shows median (white dot) and range. All career stages rated the handbook commitments as significantly important (i.e. above the grey dotted line, one-sample t-test, ** = $p < 0.001$). The importance scores from group leaders were significantly lower than lab members (Mann–Whitney U test, ** = $p < 0.001$). (**b**) Mean importance score for each commitment and career stage, sorted in ascending order based on group leaders' answers. (**c**) As in (**a**), but for the implementation rate. All career stages reported low levels of implementation, significantly below 50% in most cases (grey dotted line, one-sample t-test, ** = $p < 0.001$). The implementation rate indicated by group leaders was significantly higher than lab members (Mann–Whitney U test, * = $p < 0.05$). (**d**) As in (**b**), but for the implementation rate. (**e**) Relationship between mean importance score (x-axis) and mean implementation rate (y-axis) for each commitment as scored by group leaders. The significant correlation was captured by a robust linear fit (black, dotted 95% confidence intervals, $P<0.001$) (**f**) As in (**e**), but for postdocs ($P>0.05$). (**g**) As in (**e**), but for PhD students ($P>0.05$). (**h**) As in (**e**), but for staff ($P>0.05$). (**i**) Relationship between difference in importance score between group leaders and lab members (x-axis), and the global implementation rate across all groups (y-axis). The significant correlation was captured by a robust linear fit (black, dotted 95% confidence intervals, $P<0.001$).

The online version of this article includes the following figure supplement(s) for figure 3:

**Figure supplement 1.** Survey results were comparable across countries.

**Figure supplement 2.** Importance score split by seniority in career stage.

*supplement 1a–d*). There was a strong correlation ($R=0.67$, $P<0.001$) between implementation rate and importance, as rated by group leaders (*Figure 3e*). This suggests that group leaders prioritize commitments they perceive as more important, but it may also reflect a confirmation bias. The same correlation was also significant

when separated by country (*Figure 3—figure supplement 1e–l*).

The importance scores demonstrate that the commitments benefit both group leaders and lab members. Group leaders are accountable for writing and implementing the commitments, creating a framework to communicate key information to their group. For lab members, these commitments establish expectations and responsibilities for each role, define policies to uphold, and establish feedback mechanisms to initiate change or highlight failures.

Commitments had a higher importance rating when scored by lab members than group leaders (*Figure 3a and b*; P<0.01, Mann–Whitney U test), suggesting that group leaders systematically underestimate the importance of these commitments to their lab members. Lab members also reported lower implementation rates than group leaders (*Figure 3c and d*, P<0.05, Mann–Whitney U test). This may be because group leaders overestimate the completeness of their implementation, or that lab members are unaware of the policies – reinforcing the importance of communicating lab policy through a written medium. We observed a correlation between importance and implementation rate for group leaders (P<0.001, robust linear fit; *Figure 3e*), but not for lab members (*Figure 3f–h*), suggesting a disconnect between the actions taken by group leaders, and those that would be most valuable from the perspective of their lab. Indeed, the mean implementation rate across respondent groups was significantly correlated with the difference in

importance score between group leaders and lab members (P<0.001, robust linear fit; *Figure 3i*), suggesting group leaders prioritise policies they deem important, but not necessarily those most valued by group members.

Lab members and group leaders supported internal, rather than public, documentation of lab policies. When SAFE Labs Handbook authors did not agree on whether a commitment should be publicly or internally documented *Figure 2—figure supplement 1h*, survey respondents were asked for their input. Group leaders indicated a broad preference against public documentation (public preference ranged from 9 to 26%), while lab members were more supportive (12–75% public preference; *Figure 4—figure supplement 1*). These results suggest that while public documentation is considered valuable to some lab members, it is most important the information is documented. This informed our plans for the handbook (see Discussion).

Our survey results could be biased by sampling individuals willing to volunteer their time to complete the survey. Therefore, to test the general applicability of our results, we implemented a paid survey, asking current PhD students to rate a subset of 10 commitments (see Methods). In support of our initial survey, there was a strong correlation (P<0.001, robust linear fit) with unpaid participants when considering the relative importance across commitments (*Figure 4a*), although importance scores were marginally lower for this paid cohort (*Figure 4b*).

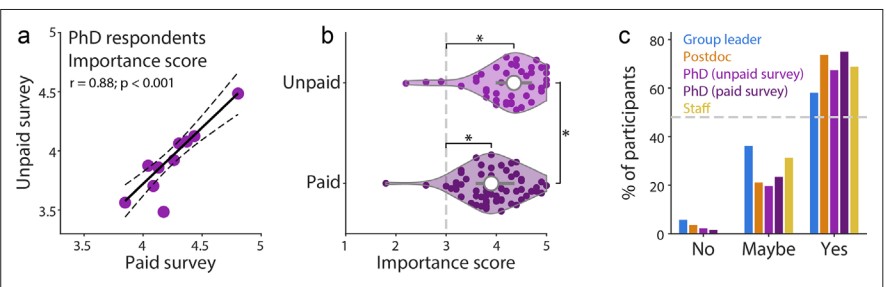

**Figure 4.** Support for the handbook is strong across all respondent groups. (**a**) Relationship between mean importance rating of 10 commitments (subset chosen for the paid survey) by PhD students from the paid and unpaid survey. The significant correlation was captured by a robust linear fit (black, dotted 95% confidence intervals, P<0.001). (**b**) Importance scores for both paid and unpaid respondents were significantly above 3 (one sample t-test), and significantly higher for unpaid than paid (Mann–Whitney U test, P<0.01). n=46 voluntary and 64 paid PhD respondents. (**c**) Fraction of respondents from each survey (paid and unpaid) in favour of implementing the handbook in their environment. For all panels, n=105 group leaders, 57 postdocs, 46 PhD students, 64 paid PhD students and 16 staff.

The online version of this article includes the following figure supplement(s) for figure 4:

**Figure supplement 1.** Support for public documentation of selected commitments.

**Figure supplement 2.** Support for handbook implementation divided by subgroups.

More than 50% of group leaders, and 70% of lab members (both unpaid and paid participants), supported full implementation of the handbook in their lab environment (*Figure 4c*, *Figure 4—figure supplement 2*). Critically, less than 6% of respondents were unsupportive of the handbook. Instead, a significant fraction of respondents indicated "maybe"—often citing institutional limitations or preferences for internal documentation (see Discussion). Given the verified importance of these commitments, this level of implementation has the potential to generate systemic change throughout the academic community.

## Discussion

The SAFE Labs Handbook is a tool that helps group leaders to create more positive and equitable environments. Importantly, the commitments in the handbook can be implemented by individual groups without institutional support. A cornerstone of the SAFE Labs approach is the recognition that each group leader has a distinct approach to management. Thus, commitments do not specify how a group should be managed, and different approaches are often equally valid, and appealing to different lab members. Moreover, we do not envision that the handbook will resolve all issues, or stop overt abuses of power (*Moss and Mahmoudi, 2021*). Rather, it will support group leaders who are committed to improving the culture in their research group.

Our survey showed that the commitments in the handbook are widely recognized as important but are implemented in a minority of labs. Despite all participant groups agreeing on the *importance* of the commitments, the implementation rate was notably low (<25%). We believe this percentage may even be an overestimate, as respondents included lab members from the authors of this document, where implementation is more probable. This gap between perceived importance and actual implementation demonstrates the value that the SAFE Labs Handbook can bring to the community. Group leaders are already convinced of the importance of these commitments, but the handbook will both increase awareness and ease implementation through example documentation.

Although implementation rate and importance were strongly correlated for group leaders, this correlation was not significant for lab members, suggesting that group leaders could better align their efforts with those of their lab members. Group leaders rated two commitments as marginally more important than lab members ('Individual Feedback' and 'PhD Committee'). Of the remaining commitments, scored higher by lab members, the largest differences were observed for 'Postdoc Funding', 'Ambitions for Roles' and 'Responsibilities'. This reveals an interesting trend, with group leaders assigning more importance to opportunities for scientific guidance, and lab members emphasizing clarity around expectations and career development.

Survey respondents reflect significant sampling biases, but correlate with the opinions of an independent cohort of paid participants. The predominance of survey respondents from the UK, France, and Italy, as well as the fields of neuroscience and cellular biology, reflect the composition of the authors of this manuscript, as well as biases in grant mechanisms and the willingness of participants to volunteer their time (see Methods). Nonetheless, our results were strongly correlated with data from a separate pool of paid participants, indicating that these results reflect opinions of the wider academic community. Nevertheless, we acknowledge that the handbook would benefit from an increased diversity of voices. Indeed, with more data, differences in the implementation rate or perceived importance across countries could serve as important indicators that changes need to be made at the institutional or national level to support group leaders.

The SAFE Labs Handbook is a dynamic resource document that will evolve through community-driven feedback and contribution. Our survey data establish the need for this tool, and the importance of the existing commitments. However, there remains significant scope for revision and expansion, both from feedback as labs implement the handbook, and from further workshops. The handbook repository includes a growing collection of real-world examples from existing implementations to help new labs develop their policies. Moreover, future SAFE Labs workshops will focus in-part on proposing changes to the handbook via the lens of a new cohort of group leaders and organisers (https://safelabs.info/).

Individual handbooks should also be considered 'living documents', and we expect both public and internal documentation to change as each group evolves. These changes should be communicated to existing lab members, and consulting lab members when updating the content of the handbook can be a valuable process. However, we encourage all group leaders to complete their initial documentation independently (indeed, this is the only option for new group leaders). We believe an honest assessment

from the group leader regarding their preferred lab policies is the most effective starting point in producing a useful handbook. All new members should be made aware of policies before joining and asked to formally acknowledge the handbook content during onboarding (e.g. providing a signature). If a member does not adhere to these values, feedback should be communicated through the established mechanisms – which are part of the existing commitments.

In addition to quantifiable metrics, our survey presented participants with the opportunity to comment on individual commitments, and on the complete handbook. We address the four most common concerns below.

### Concern 1: A commitment is covered at the institutional level

This was most common with respect to aspects of employment contracts (e.g. working hours, holiday). We now clarify that 'documenting' could involve linking an institutional policy and stating that it reflects your expectations for lab members. However, exceptions to institutional policies should be clarified (e.g. remote-working, experiments/conferences on weekends). Group leaders can provide an anonymous feedback channel for lab members to highlight any edge-cases that remain unclear.

### Concern 2: The situation is too individualistic, so documentation would be meaningless

This was most common when commitments impacted individual projects (e.g. authorship, resource management, funding of positions). We argue that this perspective (that the situation is too individualistic) also reflects an expectation that should be documented and would still be informative for lab members. We reiterate that the handbook is not prescriptive, and that a commitment would be satisfied by stating that no general lab policy exists.

### Concern 3: I do not feel comfortable publicly documenting this information

Several group leaders indicated this was the primary reason that they would 'maybe' implement the handbook. As group leaders, we understand this perspective, particularly given that publicly documented material in academia is atypical. We maintain that public documentation is a valuable resource for prospective lab members, and this in turn increases the likelihood of group leaders recruiting suitable candidates

for positions. This is evidenced by the higher enthusiasm for public documentation amongst lab members. However, to encourage broader adoption of the handbook, we have removed the requirement for public documentation from many commitments.

### Concern 4: Implementing these commitments is too much work

This criticism referenced two features: meetings and documentation. Although the handbook encourages group leaders to document regular meetings and expectations, complete implementation of the handbook only requires three annual meetings (normalising failure, lab-wide feedback, and core-skills development). This would minimally impact the annual calendar of most labs. Conversely, the documentation required represents a significant workload, exceeding the norm in academia. However, we envision this time will be recouped by pre-empting questions from lab members and reducing conflicts that arise from expectation mismatch or inter-personal differences. We provide concrete templates and suggestions for each commitment to simplify implementation. Furthermore, we have a growing list of community examples from group leaders that have implemented the handbook (already a minimum of two for each commitment). We anticipate implementation times to continue shrinking as this repository expands, and group leaders can select an example that mirrors their own policies.

## Next steps

We envision the SAFE Labs Handbook as a critical tool for group leaders, as well as an opportunity to advertise their commitment to creating more positive and equitable academic environments. We acknowledge that the handbook is not sufficient to create a positive lab culture, and that this relies on sustained efforts from both group leaders and lab members. However, if expectations are clear to all lab members at the earliest opportunity, it creates a valuable foundation from which future challenges and conflicts can be addressed. We expect the handbook to be most effective in an environment that encourages open and honest bidirectional feedback, and the commitments insist on establishing dedicated meetings for this purpose. It is not possible to eliminate difficulties in lab members confronting group leaders that fail to adhere to their own handbook, but we include suggestions in the handbook for reducing this barrier (e.g. an anonymous feedback channel).

To maximise future engagement with both the handbook and future workshops, we hope to disseminate information more effectively. Avenues could include institutional welcome packages for new group leaders, early career funding schemes, and professional networks such as EMBO. Engagement is also encouraged through our dedicated mailing list and Github discussions (https://github.com/SAFE-Labs-Docs/Lab-Handbook/discussions). We maintain a list of SAFE Labs on our website (https://safelabs.info/home/the-labs/). Labs that want to be featured on this page are required to submit a request, together with a link to their publicly documented handbook or policies. These submissions are briefly scrutinised before labs are added to the community list, and the lab is encouraged to feature the lab logo on their website in recognition of their implementation. These logos are considered indicators of engagement, like open science badges, and advertise to prospective scientists that the group leader places importance on lab culture. A more formal certification system could be developed in the future for use as a key performance indicator at the institutional level and could be recognised by funders.

## Methods

### Application process for workshop

The workshop was advertised through a combination of social media posts and targeted emails to newly appointed group leaders across Europe, within three years of starting their lab through awarded grants (e.g. the ERC starting grant). The application included 14 questions, spanning demographics, motivational statements, and potential discussion topics (*Supplementary file 1*). The data and ideas from the application were analysed to shape the SAFE Labs workshop *Figure 2—figure supplement 1a, b*. Nine participants were selected from a total of 21 applicants based on blind-scoring by the organisers, weighted by the need to maximise diversity across countries and scientific fields.

### Workshop format and handbook preparation

The workshop comprised six sessions, based on topics highlighted by applicants. Each session had three phases: an introductory presentation, small group discussions, and a joint roundtable. The introductory presentation was prepared in advance and delivered by one of the applicants. For group discussions, participants were randomly divided into three subgroups, each with a randomly assigned group leader and note-taker, to discuss three questions on the current topic: What are the biggest issues facing new labs? What can new PIs do to tackle/prevent these issues? What could institutions do to support new PIs? During the joint roundtable, subgroups convened to summarise and discuss the main conclusions from each subgroup.

### Workshop exit survey

After the workshop, the organizers solicited feedback from attendees through an anonymous exit survey, consisting of 22 questions. Participants were asked to indicate their level of agreement with selected statements, to score the session topics and formats, and the logistical aspects of the meeting (*Supplementary file 2*; *Figure 2—figure supplement 1e–h*).

### Voluntary survey

The voluntary survey was designed to ascertain the level of support for the SAFE Labs Handbook from the research community. The survey was advertised through social media, by soliciting scientific organisations and publications, and direct emails to grant awardees and institutional contacts. In particular, the survey was emailed to recipients of the ERC starting grant from the year 2021–2024, and other national early career grants (e.g. Italian PRIN under 40, Human Technopole Early Career Fellowship). The survey respondents therefore reflect a combination of biases, including the professional and social networks of the authors, the inequalities systemic to grant awardees, and those willing to invest significant time to this initiative.

The survey consisted of 106 questions (33 optional) to evaluate the handbook (*Figures 3 and 4*), which was made available via Github (see Resources). For each of the 30 commitments, participants were asked to rate its importance on a scale from 1 (useless) to 5 (critical); to indicate whether it had already been implemented in their research environment; and to provide further optional comments. For a subset of commitments (7/30) participants were asked whether the object of the commitment should be publicly or internally documented. At the conclusion, participants were asked if they would implement, or want their group leader to implement, the Handbook (*Supplementary file 3*).

A pilot survey was trialled with the 2024 SAFE Labs workshop participants. This served to revise elements of the handbook and survey

that might have caused confusion prior to public release, and identified commitments without a clear consensus for public vs internal documentation, to be further interrogated through the public survey (7 commitments; *Figure 2—figure supplement 1h*).

### Paid survey

The paid survey was implemented through Prolific software (https://www.prolific.com/). To maximise similarity to the voluntary data, participants were limited to biosciences PhD students from countries that featured prominently in the volunteer sample. Costs were minimised by selecting 10 of the 30 commitments (with an additional attention-check) that received a range of scores by voluntary PhD students (*Supplementary file 4*). Participants were asked to score the importance of these commitments, but did not receive as much background information about the handbook, in order to limit costs. A total of 64 paid respondents were included in analyses (*Figure 4*).

### Data analysis and statistics

Survey data were pre-processed to classify respondents to standardised research roles and categories. For research roles we used the following classification:

| Standardized role | Open survey answer |
|---|---|
| Group Leader | Assistant Professor, Principal Investigator, Project Leader, Group Leader |
| Postdoc | Researcher, Newly Graduated as PhD, Research Fellow, Research Scientist, Postdoc |
| PhD Student | Intern, Master Student, PhD Student |
| Staff | Research Engineer, Research Assistant, Technician, Lab manager |

For research fields we used the following classification:

| Standardized field | Open survey answer |
|---|---|
| Neuro & Behaviour | Neuroscience, Neurobiology, Neuroscience and Behavioural Sciences, Psychology |

*Continued on next page*

| Standardized field | Open survey answer |
|---|---|
| Cell & Molecular | Biophysics, Cell Physiology, Molecular Biology and Genetics, Functional Genomics, Cell Biology and Cellular Signalling |
| Cancer | Cancer Biology, Molecular Oncology, Cancer Biology |
| Developmental | Developmental Biology and Stem cells |
| Immuno & Microbiology | Microbiology, Molecular Biology, Microbiology and Virology, Microbial Evolutionary Ecology, Parasitology, Immunology and Infectious Diseases |
| Synthetic & Structural | Chemical engineering, Synthetic, Chemical and Synthetic Biology, Biochemistry, Biotechnology, Functional morphology, Biochemistry and Structural Biology, Chemistry |
| Ecology & Evolution | Ecology and Evolutionary Biology |
| Plants | Plant Biology |
| Bioinformatics | Systems Biology and Bioinformatics |
| Other | Medical Imaging, Optics, Imaging Physics |

Data were then analysed in MATLAB with custom code. Differences in responses across categories were assessed using the Kruskall–Wallis test or Mann–Whitney U test. The relation between importance and established score was fit with robust linear regression.

### Acknowledgements

The SAFE Lab initiative was funded by the UCL Global Engagement Fund (to SB, LM, LFR and PC). The workshop was hosted under the patronage of Scuola Normale Superiore and the sponsorship of Professor Franco Ligabue. We would also like to thank the 224 participants who volunteered their time to complete the survey. We thank Sara Saab (Prolific) for facilitating the paid survey, and Prolific for providing credit to partially cover these costs.

**Erika Donà** is in the Institute of Neuroscience, Consiglio Nazionale delle Ricerche (CNR), Vedano al Lambro, Italy

https://orcid.org/0000-0001-9268-3619

**James M Gahan** is in the Centre for Chromosome Biology, School of Biological and Chemical Sciences, University of Galway, Galway, Ireland

https://orcid.org/0000-0001-7043-4873

Petrina Lau is in the Department of Psychiatry and the Gerald Choa Neuroscience Institute, The Chinese University of Hong Kong, Shatin, Hong Kong

https://orcid.org/0000-0003-2257-2344

Jana Jeschke is in the Institut Jules Bordet, Université libre de Bruxelles, Brussels, Belgium

Torben Ott is in the Institute of Biology and Bernstein Center for Computational Neuroscience Berlin, Humboldt University of Berlin, Berlin, Germany

https://orcid.org/0000-0002-4168-2832

Katja Reinhard is in the Neuroscience Department, Scuola Internazionale Superiore di Studi Avanzati, Trieste, Italy

https://orcid.org/0000-0002-8719-7445

Chiara Sinigaglia is at the Biologie Intégrative des Organismes Marins (BIOM), CNRS and Sorbonne Université, Banyuls-sur-Mer, France

https://orcid.org/0000-0002-7195-7091

Jorien L Treur is in the Department of Psychiatry, Amsterdam UMC, University of Amsterdam, Amsterdam, Netherlands

Thomas Vogl is in the Center for Cancer Research, Medical University of Vienna, Vienna, Austria

Stephane Bugeon is at the Institut de Neurobiologie de la Méditerranée (INMED), INSERM and Aix Marseille Université, Marseille, France

stephane.bugeon@inserm.fr

https://orcid.org/0000-0002-5703-4848

Letizia Mariotti is in the Institute of Neuroscience, Consiglio Nazionale delle Ricerche (CNR), Padova, Italy

letizia.mariotti@cnr.it

https://orcid.org/0000-0002-6749-1006

L Federico Rossi is in the Center for Neuroscience and Cognitive Systems, Istituto Italiano di Tecnologia, Rovereto, Italy

federico.rossi@iit.it

https://orcid.org/0000-0001-5831-4860

Philip Coen is in the Cell and Developmental Biology Department, University College London, London, United Kingdom

p.coen@ucl.ac.uk

https://orcid.org/0000-0003-1495-1061

Author contributions: Erika Donà, Conceptualization, Writing – review and editing; James M Gahan, Conceptualization, Writing – review and editing; Petrina Lau, Conceptualization, Writing – review and editing; Jana Jeschke, Conceptualization, Writing – review and editing; Torben Ott, Conceptualization, Writing – review and editing; Katja Reinhard, Conceptualization, Writing – review and editing; Chiara Sinigaglia, Conceptualization, Writing – review and editing; Jorien L Treur, Conceptualization, Writing – review and editing; Thomas Vogl, Conceptualization, Writing – review and editing; Stephane Bugeon, Conceptualization, Data curation, Formal analysis, Funding acquisition, Investigation, Visualization, Methodology, Writing – original draft, Project administration, Writing – review and editing; Letizia Mariotti, Conceptualization, Data curation, Formal analysis, Funding acquisition, Investigation, Visualization, Methodology, Writing – original draft, Project administration, Writing – review and editing; L Federico Rossi, Conceptualization, Data curation, Formal analysis, Funding acquisition, Investigation, Visualization, Methodology, Writing – original draft, Project administration, Writing – review and editing; Philip Coen, Conceptualization, Data curation, Formal analysis, Funding acquisition, Investigation, Visualization, Methodology, Writing – original draft, Project administration, Writing – review and editing

Competing interests: The authors declare that no competing interests exist.

Ethics: Human subjects: The paper includes anonymised survey data from members of the academic community.

## Funding

| Funder | Grant reference number | Author |
| --- | --- | --- |
| University College London | Global Engagement Award | Stephane Bugeon Letizia Mariotti L Federico Rossi Philip Coen |

The funders had no role in study design, data collection and interpretation, or the decision to submit the work for publication.

Decision letter and Author response
Decision letter https://doi.org/10.7554/eLife.108853.sa1
Author response https://doi.org/10.7554/eLife.108853.sa2

## Additional files

### Supplementary files
Supplementary file 1. SAFE Labs 2024 application form data.

Supplementary file 2. SAFE Labs 2024 exit survey data.

Supplementary file 3. Handbook public survey data.

Supplementary file 4. Prolific paid survey data.

MDAR checklist

### Data availability
All data generated or analysed during this study are included in the accompanying Git repository (https://github.com/SAFE-Labs-Docs/2025_Paper; copy archived at *Coen, 2025*). All data, analyses, and updated versions of the handbook will remain available at the SAFE Labs Handbook website (https://safelabs.info/home/safe-labs-handbook/) and the SAFE Labs

Handbook GitHub repository (https://github.com/SAFE-Labs-Docs/Lab-Handbook; commit 971f8bf at time of submission).

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
