## [Decision Letter]

**Decision letter after peer review:**

Thank you for submitting your article "The SAFE Labs Handbook: Community-Driven Commitments for Group Leaders to Improve Lab Culture" to *eLife* for consideration as a Feature Article. Your article has been reviewed by three peer reviewers, and there reports are below. The following reviewer has agreed to reveal their identity: Benjamin Tendler (Reviewer #3).

General Assessments

*Reviewer #1:*

This is an important article that provides concrete action items and resources for group leaders who are seeking to document their lab policies in a handbook. This is a much-needed initiative with the potential to improve transparency in academia, which in turn is important for improving lab culture and mental well-being. Its strengths include: (i) data on the perceived importance and implementation of individual commitments and on willingness to implement lab handbooks; (ii) concrete advice, templates, and an easily accessible starting point to what can be a daunting task of creating a handbook; (iii) excellent acknowledgement of limitations in the approach and biases inherent to the sampling of workshop and survey participants; and (iv) a clearly written article with links to resources that the community can use. The main limitations revolve around an increased need for discussion of several issues, summarized below.

*Reviewer #2:*

I had come across the SAFE labs initiative before, and read this work with great pleasure. I strongly agree with the author that bottom-up, community initiatives are a key ingredient of academic culture change, and this work is a fantastic step in that direction. Given *eLife*'s commitment to community and research culture, I think this would be the ideal outlet for publication.

This article is a bit of a mix between a general feature (to showcase the workshop and handbook itself) and research (surveys to draw more quantitative conclusions). While I think the latter is great to have, it is the former that I believe will have the greatest impact through wider communication about the existence of the handbook. I have a few suggestions that the authors may consider (see below).*Reviewer #3:*

In this work, Donà and colleagues present the SAFE Labs Handbook, an evidence-based research culture initiative actionable by research groups. The SAFE Labs Handbooks initiative proposes the adoption of a document comprising of 30 community-driven 'commitments' designed to improve the way research groups operate. The recommendations are backed up by community evidence (taken in the form of surveys), designed to be easily actionable, and are non-prescriptive to incorporate different viewpoints.

Establishing the 30 commitments utilised a carefully considered methodology. The researchers have gone to considerable effort to quantitatively evaluate their recommendations, demonstrating that their commitments are perceived as important at all levels of a research group whilst simultaneously poorly described/documented in lab policy. The number of survey respondents (particularly from group leaders) is impressive, and also helps extract differences in how researchers at different career stages interpret the relative importance and implementation of different policies. From my perspective, I frame the commitments as a distillation of previous lab handbook initiatives designed to improve research culture, sourcing evidence from communities (as opposed to an individual group), and describing a direct set of recommendations that can be more rapidly implemented without institutional support.

Whilst the authors go to great length to describe how they established their commitments, the aim of their work is not simply to establish a template, but to launch an initiative promoting the adoption of SAFE handbooks in the wider research community. A key area that should be addressed is how to approach writing the handbook, who it is for, and how it is incorporated into a research lab. Below I provide some comments focused on these themes.

Essential Revisions:

*Reviewer #1:*

[1] The Introduction states that "academic environments are associated with impaired mental health, poor work-life balance, and disparities in pay and career progression". It is important to clarify what the comparison group is. Impaired mental health compared to what group? Poor work-life balance compared to what group? Is the comparison group matched on important demographic features, and differ only in the presence vs. absence of an academic work environment? Or does this claim rest on self-reported ratings of mental health and work-life balance without a matched comparison group? Likewise, it seems important to clarify whether the disparities in pay and career progression are larger in academia vs. other settings.

[2] It is stated that "existing resources provide general advice rather than actionable steps that can be verifiably implemented". Yet, some of the cited papers include links to publicly available lab policies that touch on many of the SAFE Labs Handbook Commitments, and these papers invite group leaders to follow their example by creating their own lab policies using those publicly available ones as a template. I definitely see the value of the SAFE Labs Handbook and the list of commitments, but the statements that prior work in this space has been more general advice rather than actionable steps does not seem accurate.

[3] The paper notes a disconnect between actions taken by group leaders and those perceived to be most important by lab members. It further notes that increased divergence between group leaders and lab members was correlated with a higher global importance score (Figure 3i). But Figure 3i seems to suggest that the actions with the most divergence (-0.5 or lower on the y-axis) are relatively *less* important. Actions that are more important (>4.5 on the x-axis) have values closer to 0 on the y-axis (e.g., -0.2), indicating less divergence between group leaders and lab members. Because most of the points in that figure lie below 0 on the y-axis, more divergence reflects *lower*, more negative, y-axis values, which are related to *lower* importance scores on the x-axis. This seems to contradict the statement in the paper that increased divergence is correlated with higher global importance. Can the authors clarify where the misunderstanding is?

[4] There are two additional discussion points that I think are worth including in the paper. First is the difference between implementing the suggested practices and actually having a positive lab culture. Transparently communicating expectations does not necessarily lead to a positive lab environment. In addition to fair and equitable expectations, a positive lab environment requires sustained efforts on the part of the group leader and the lab members. It might be advisable to mention that this lab handbook approach should be the first step, and that it may be necessary but is certainly not sufficient for a supportive environment. A second, related, issue is that of accountability. I appreciate the recommendation for annual lab-wide feedback sessions and mechanisms for sharing concerns and conflicts. But it can be very difficult for lab members to confront their group leader when the group leader is not following their own stated policies. This challenge is almost inevitable because of power dynamics and how much lab members depend on their group leader for advancement. It may be worth discussing how it can be very difficult to hold group leaders accountable and ensure that they follow their stated policies. It would also be valuable to include recommendations for how this can be approached in a way that protects lab members and eases the discomfort and anxieties associated with giving their group leader negative feedback.*Reviewer #2:*

[5] My first one is to speculate about future ways in which the potential of this resource could be maximized. The website will spread through subfields by word-of-mouth, some people will see the article/preprint, and a low number may attend the workshop itself as a new PI. Would there be other ways to include this resource in e.g. welcome packages at institutions? Through funders who award early career grants? Through e.g. FENS, EMBO, SfN, NeuroMatch, NewPI Slack? I do not expect the authors to carry out all this work in community building, but a vision for the way forward would help in identifying where new cohorts of the workshop could build on further anchoring these practices in the community.

[6] The second one is to zoom in a bit to the items that have the highest discrepancy between PIs and lab members, from figure 3i. Which topics would need most attention or additional discussion?*Reviewer #3:*

[7] Writing a SAFE Handbook

The survey findings identify an increased interpreted level of policy implementation for group leaders versus members (Figure 3c), suggesting that policies are being miscommunicated by group leaders to group members. In addition, the findings identify increased perceived importance of several topics from group members versus leaders (Figure 3a), suggesting that the voices of group members are not being effectively heard. This evidence suggests that a SAFE Handbook should be written collectively as a group exercise rather than a top-down approach, in agreement with previous handbook initiatives that strongly advocated for writing as a group exercise (Tendler et al. 2023), or findings highlighting the gap of interpretation of expectations (Wellcome Trust 2020).

The manuscript does not present any information about how to approach writing a SAFE handbook beyond hinting as a group writing exercise in 'Concern 4'. The authors should add a clear description of this to ensure that the benefits of the document are maximised, and do not risk parts of the initiative boiling down to a template copy-and-paste box ticking exercise by a group leader without broader thought and consideration of lab viewpoints.

[8] Who is the Handbook for?

The writing of the manuscript and SAFE Handbook website suggests that the thirty commitments are for group leaders. However, I would argue that many of the themes (e.g. code of conduct) are the responsibility of everybody in the lab, and require an explicit commitment from individual group members. Please describe and provide justification as to who is accountable for the contents of the lab handbook. If the expectation is that ensuring the commitments are adhered to is the responsibility of the entire group, recommendations for achieving group buy-in.

[9] How is the Handbook integrated into the lab?

Please describe more explicitly what the expectations are for the established handbook in the lab. Specifically, (1) the expectations surrounding group members reading and committing to the handbook, (2) the recommended process if somebody is not living up to the values expressed in the handbook, (3) expectations for the handbook content to be reviewed and updated (e.g. on an annual basis as a 'living document').

---

## [Author Response]

Essential Revisions:Reviewer #1:[1] The Introduction states that "academic environments are associated with impaired mental health, poor work-life balance, and disparities in pay and career progression". It is important to clarify what the comparison group is. Impaired mental health compared to what group? Poor work-life balance compared to what group? Is the comparison group matched on important demographic features, and differ only in the presence vs. absence of an academic work environment? Or does this claim rest on self-reported ratings of mental health and work-life balance without a matched comparison group? Likewise, it seems important to clarify whether the disparities in pay and career progression are larger in academia vs. other settings.

We agree with the reviewer and have now corrected this omission in the manuscript (Intro, ∼ Para 1). We now make it clear that these studies lack comparisons to demographically matched control groups, but that they still suggest systemic issues within academia that need to be addressed.

[2] It is stated that "existing resources provide general advice rather than actionable steps that can be verifiably implemented". Yet, some of the cited papers include links to publicly available lab policies that touch on many of the SAFE Labs Handbook Commitments, and these papers invite group leaders to follow their example by creating their own lab policies using those publicly available ones as a template. I definitely see the value of the SAFE Labs Handbook and the list of commitments, but the statements that prior work in this space has been more general advice rather than actionable steps does not seem accurate.

We agree with the reviewer that there are existing resources providing both general advice and specific examples. These resources are incredibly valuable, and we agree that our previous statement unintentionally minimised this contribution. However, as the reviewer points out, these resources typically consist of a single, actionable step: using an example handbook to help create a personalised one. We feel that the breakdown of this process into separate, actionable, commitments will make this process more manageable and more universally applicable. We now specify this distinction in the manuscript (Intro, ∼Para 2).

[3] The paper notes a disconnect between actions taken by group leaders and those perceived to be most important by lab members. It further notes that increased divergence between group leaders and lab members was correlated with a higher global importance score (Figure 3i). But Figure 3i seems to suggest that the actions with the most divergence (-0.5 or lower on the y-axis) are relatively *less* important. Actions that are more important (>4.5 on the x-axis) have values closer to 0 on the y-axis (e.g., -0.2), indicating less divergence between group leaders and lab members. Because most of the points in that figure lie below 0 on the y-axis, more divergence reflects *lower*, more negative, y-axis values, which are related to *lower* importance scores on the x-axis. This seems to contradict the statement in the paper that increased divergence is correlated with higher global importance. Can the authors clarify where the misunderstanding is?

We thank the reviewer for raising this issue: our description of the correlation in the previous figure was mis-stated. Further, the effect was small and not significant when evaluated with Pearson correlation, the metric used in panels 3e-h (see minor point, Reviewer 1.c), we have therefore replaced Figure3i with a more informative analysis, capturing a more substantial effect.

The new Figure 3i shows that the difference in importance-score between group leaders and group members correlates with the mean implementation rate across groups. Thus, implementation is lower for commitments that group members deem more important than group leaders; in other words, group leaders prioritise policies they deem important over those that may be more valued by their group members. We have corrected the figure, legend, and text (Results of a Community Survey, ∼ Para 4).

[4] There are two additional discussion points that I think are worth including in the paper. First is the difference between implementing the suggested practices and actually having a positive lab culture. Transparently communicating expectations does not necessarily lead to a positive lab environment. In addition to fair and equitable expectations, a positive lab environment requires sustained efforts on the part of the group leader and the lab members. It might be advisable to mention that this lab handbook approach should be the first step, and that it may be necessary but is certainly not sufficient for a supportive environment. A second, related, issue is that of accountability. I appreciate the recommendation for annual lab-wide feedback sessions and mechanisms for sharing concerns and conflicts. But it can be very difficult for lab members to confront their group leader when the group leader is not following their own stated policies. This challenge is almost inevitable because of power dynamics and how much lab members depend on their group leader for advancement. It may be worth discussing how it can be very difficult to hold group leaders accountable and ensure that they follow their stated policies. It would also be valuable to include recommendations for how this can be approached in a way that protects lab members and eases the discomfort and anxieties associated with giving their group leader negative feedback.

We agree with the reviewer on both these points. We have now re-emphasised the limits of the handbook, including the difficulties in providing feedback to group leaders (Discussion, ∼ Para 12). We also highlight that the handbook includes suggestions for reducing the barrier to providing feedback to group leaders, and include one example in the paper (maintaining an anonymous feedback channel, Discussion, ∼ Para 12).

Reviewer #2:[5] My first one is to speculate about future ways in which the potential of this resource could be maximized. The website will spread through subfields by word-of-mouth, some people will see the article/preprint, and a low number may attend the workshop itself as a new PI. Would there be other ways to include this resource in e.g. welcome packages at institutions? Through funders who award early career grants? Through e.g. FENS, EMBO, SfN, NeuroMatch, NewPI Slack? I do not expect the authors to carry out all this work in community building, but a vision for the way forward would help in identifying where new cohorts of the workshop could build on further anchoring these practices in the community.

We agree with these points. We have expanded our discussion to touch on these points (∼ Discussion, ∼ Final Para). We further highlight our future plans, and the available mailing list and github discussions for community engagement. We are committed to continuing the SAFE Labs Website and repository as this initiative evolves.

[6] The second one is to zoom in a bit to the items that have the highest discrepancy between PIs and lab members, from figure 3i. Which topics would need most attention or additional discussion?

This is an excellent suggestion, and we now highlight some points of discrepancy in the discussion. This revealed a suggestive difference, where group leaders felt scientific guidance was more important while lab members highlighted the importance of clear expectations and career-development goals (Discussion, ∼Para 3).

Reviewer #3:[7] Writing a SAFE HandbookThe survey findings identify an increased interpreted level of policy implementation for group leaders versus members (Figure 3c), suggesting that policies are being miscommunicated by group leaders to group members. In addition, the findings identify increased perceived importance of several topics from group members versus leaders (Figure 3a), suggesting that the voices of group members are not being effectively heard. This evidence suggests that a SAFE Handbook should be written collectively as a group exercise rather than a top-down approach, in agreement with previous handbook initiatives that strongly advocated for writing as a group exercise (Tendler et al. 2023), or findings highlighting the gap of interpretation of expectations (Wellcome Trust 2020).The manuscript does not present any information about how to approach writing a SAFE handbook beyond hinting as a group writing exercise in 'Concern 4'. The authors should add a clear description of this to ensure that the benefits of the document are maximised, and do not risk parts of the initiative boiling down to a template copy-and-paste box ticking exercise by a group leader without broader thought and consideration of lab viewpoints.

We thank the reviewer for highlighting a significant omission from the manuscript. We agree that feedback from lab members is critical, and that the Handbook should be regularly reviewed and edited in conversation with lab members (Discussion, ∼ Para 6). However, we do not believe the initial handbook should be written as a group exercise for the following reasons:

New group leaders do not have the facility to collectively write their handbook, but we do not want to intimate that this process should be delayed until the lab is larger. Handbooks should be completed before the first lab member joins to provide them with a clear impression of the expectations of the group leader.

Commitments in the Handbook are designed to minimise expectation mismatch between lab members (both prospective and current) and group leaders, with the acknowledgement that each group leader is different. We believe an honest assessment from the group leader regarding their preferred lab policies is the most effective starting point in producing a useful Handbook, rather than deferring this responsibility to a collective effort. After the Handbook is written, feedback from lab members is extremely valuable, and if suggestions are not incorporated, the reasoning behind this should be explained.

The group leader is (usually) the only “constant” presence in the lab, while the other personalities will change over time. Therefore, we believe it is appropriate for the group leader to have a driving role in setting the expectations for their group rather than relying on lab members (likely the more vocal members).

We therefore suggest that documentation be written by the group leader in the first instance (Discussion, ∼ Para 6).

[8] Who is the Handbook for?The writing of the manuscript and SAFE Handbook website suggests that the thirty commitments are for group leaders. However, I would argue that many of the themes (e.g. code of conduct) are the responsibility of everybody in the lab, and require an explicit commitment from individual group members. Please describe and provide justification as to who is accountable for the contents of the lab handbook. If the expectation is that ensuring the commitments are adhered to is the responsibility of the entire group, recommendations for achieving group buy-in.

The reviewer makes an excellent point, echoing feedback that we also received from our survey, which we should have addressed more plainly. We now clarify (Results of a Community Survey, ∼ Para 3) that the Handbook is for the benefit of both group leaders and lab members. Group leaders are accountable for writing an implementing the commitments, creating a framework to communicate key information to their group. For lab members, these commitments establish expectations and responsibilities for each lab role, define policies to uphold, and establish feedback mechanisms to initiate change or highlight failures. To promote this framework, we now specify that the onboarding procedure needs to include reading and acknowledging the lab handbook by the lab member.

[9] How is the Handbook integrated into the lab?Please describe more explicitly what the expectations are for the established handbook in the lab. Specifically, (1) the expectations surrounding group members reading and committing to the handbook, (2) the recommended process if somebody is not living up to the values expressed in the handbook, (3) expectations for the handbook content to be reviewed and updated (e.g. on an annual basis as a 'living document').

These are excellent suggestions, and we now specifically address them in the revision (Discussion, ∼Para 6).